# Formation of Metal-Oxide Nanocomposites with Highly Dispersed Co Particles from a Co-Zr Powder Blend by Mechanical Alloying and Hydrogen Treatment

**DOI:** 10.3390/ma16031074

**Published:** 2023-01-26

**Authors:** Ilya Yakovlev, Serguei Tikhov, Evgeny Gerasimov, Tatiana Kardash, Konstantin Valeev, Aleksei Salanov, Yurii Chesalov, Olga Lapina, Oleg Lomovskii, Dina Dudina

**Affiliations:** 1Boreskov Institute of Catalysis SB RAS, Lavrentyev Ave. 5, 630090 Novosibirsk, Russia; 2Institute of Solid State Chemistry and Mechanochemistry SB RAS, Kutateladze Str. 18, 630090 Novosibirsk, Russia; 3Lavrentyev Institute of Hydrodynamics SB RAS, Lavrentyev Ave. 15, 630090 Novosibirsk, Russia

**Keywords:** mechanical alloying, hydrogenation, Zr-Co powder, metal oxide nanocomposite, catalyst

## Abstract

The use of metal powders produced by mechanical treatment in various fields, such as catalysis or gas absorption, is often limited by the low specific surface area of the resulting particles. One of the possible solutions for increasing the particle fineness is hydrogen treatment; however, its effect on the structure of mechanically treated powders remains unexplored. In this work, for the first time, a metal-oxide nanocomposite powder was produced by mechanical alloying (MA) in a high-energy planetary ball mill from commercial powders of Zr and Co in the atomic ratio Co:Zr = 53:47 in an inert atmosphere, followed by high-pressure hydrogenation at room temperature. The initial powders and products of alloying and hydrogenation were studied by XRD, ^59^Co Internal Field NMR, SEM, and HRTEM microscopy with EDX mapping, as well as Raman spectroscopy. MA resulted in significant amorphization of the powders, as well as extensive oxidation of zirconium by water according to the so-called “Fukushima effect”. Moreover, an increase in *hcp* Co sites was observed. ^59^Co IF NMR spectra revealed the formation of magnetically single-domain cobalt particles after hydrogenation. The crystallite sizes remained unchanged, which was not observed earlier. The pulverization of Co and an increase in *hcp* Co sites made this nanocomposite suitable for the synthesis of promising Fischer–Tropsch catalysts.

## 1. Introduction

Cobalt–zirconium alloys are widely used as hydrogen storage materials, active gas adsorbents under high vacuum conditions [1,2,3,4,5,6,7,8,9] and magnetic materials [9]. The properties of these alloys significantly depend on the phase and chemical composition, crystal structure, concentration of defects and particle size [10,11,12]. The mechanochemical synthesis is often used to alter the phase composition, increase the defect concentrations, induce amorphization and reduce the particle size of the powder alloys [11,12,13,14,15,16]. In most publications, systems with an excess of zirconium were synthesized and studied. At the same time, there are reports on systems with a Co:Zr ratio equal to 1:1 [16] and systems with an excess of cobalt [15]. The mechanical alloying (MA) process is often followed by annealing. In ref. [16], the process of MA of a Co-Zr system under various conditions was studied; no annealing was applied to the products. The literature data are summarized in Table 1.

Me-Zr alloys are promising materials for the catalytic reaction of CO hydrogenation [17,18,19]. MA of metal powders is achieved through high-energy ball milling. This process does not necessarily produce particles with a high specific surface area. Consequently, additional steps in the synthesis aimed at further dispersing the obtained particles are often taken. One of the possible approaches (or rather, a group of approaches) used for particle dispersion is their treatment with gaseous hydrogen under different temperatures and pressures, referred to as hydrogen pulverization. The nature of this process lies in the extreme expansion of crystal lattices after absorption of hydrogen. This has a pronounced effect in heterogeneous systems (such as mechanically alloyed powders), in which phases with different lattice expansion rates are present. A preliminary treatment in hydrogen was shown to improve the catalytic properties of the mechanically alloyed systems [20,21]. However, the detailed spectral and microstructural characteristics of the Co-Zr alloys before and after hydrogenation were not studied. 

The technique of ^59^Co Internal Field NMR spectroscopy is a powerful tool for studying systems containing cobalt in the metallic state. The occurrence of an NMR signal in cobalt metal particles is associated with their ferromagnetism, namely, with the presence of a local (internal) magnetic field at the location of the Co nucleus, which reaches values of 20–21 T in *fcc* and *hcp* cobalt structures [22,23]. The dependence of the local magnetic field on the crystalline (*fcc*, *hcp* packing) and magnetic (single- and multi-domain particles) structures makes it possible to characterize the structure of metallic cobalt particles in the sample [24,25,26,27,28]. This method also makes it possible to characterize the intermetallic compounds and solid solutions of cobalt due to the dependence of the local magnetic field on the number of guest atoms in the nearest environment of cobalt [29,30].

As applied to the binary Co-Zr system, this method was used to study the structure of multilayer systems of the …/Co/Zr/Co/… type with different metal ratios and layer thicknesses from 5 to 80 nm obtained by radio frequency sputtering [31,32]. In multilayer films obtained by this method, an amorphous alloy layer with an approximate composition of Co_80_Zr_20_ appeared between the layers of metallic Co and Zr, and the inclusion of individual Zr atoms in the local environment of Co in the region of grain boundaries was also observed. In both cases, an additive decrease in the resonant frequency relative to metallic cobalt (resonant frequency 213–219 MHz) by 30 MHz for each Zr atom located in the local environment of Co was observed due to the absence of an intrinsic magnetic moment of the Zr sites.

In this work, we studied the effect of MA and subsequent low-temperature high-pressure hydrogen pulverization on the structural and spectral properties of a cobalt-zirconium powder alloy with a Co:Zr atomic ratio of approximately 1:1. The conventional techniques (powder X-ray diffraction, electron microscopy with EDX and Raman spectroscopy) were used together with ^59^Co Internal Field NMR spectroscopy to characterize the alloys. ^59^Co Internal Field NMR spectroscopy has proven to be an efficient characterization tool for systems containing Co in the metallic state and ferromagnetic intermetallic Co compounds. It allows investigation of the crystal and magnetic structure of Co particles, describes the particle size distribution and detects the presence of guest atoms in the nearest environment of Co nuclei. There are no data in the literature indicating the application of this technique to studying Co-Zr mixtures that have undergone hydrogen pulverization.

## 2. Materials and Methods

### 2.1. Sample Preparation

A zirconium powder (PZrK-1, JSC “Donetsk chemical-metallurgical plant”) and an electrolytic cobalt powder (PK-1u, GC “Metal-Energo Holding”, Yekaterinburg, Russia) were used as the starting materials. The powders were preliminary mixed in a Co:Zr atomic ratio of 53:47. MA was carried out in an argon atmosphere using an AGO-3 planetary ball mill (Institute of Solid State Chemistry and Mechanochemistry SB RAS, Novosibirsk, Russia [33]). The mill is equipped with a water cooling system, steel jars and balls. The maximum acceleration of milling bodies was 60 g (600 m^2^/s), the carrier rotation speed was 800 rpm. Steel balls with a diameter of 5 mm were used. The ball:powder ratio was 1:10. First, the powders were processed at an acceleration of 20 g for 6 min, then at an acceleration of 60 g for 12 min. After cooling for 12 h, the vials were opened, and the resulting powder was removed.

The samples were treated with hydrogen using a high-pressure setup (Autoclave Engineers, Erie, PA, USA) in a periodic action EZE-Seal stainless steel reactor with a volume of 300 mL. The reactor was equipped with temperature and pressure sensors, as well as a system for automatic temperature and pressure control. First, the loaded reactor was purged with a N_2_ stream (200 cm^3^/min) under atmospheric pressure followed by a H_2_ purge at atmospheric pressure (200 cm^3^/min) for several minutes. Then, hydrogen at a pressure of 5.0 MPa and room temperature was introduced into the reactor volume. The samples were treated under these conditions for 24 h. After the high-pressure hydrogen treatment, the pressure was gradually decreased to 1.8–2.5 MPa, and the reactor was purged with nitrogen for 4 h in order to passivate the sample, after which the reactor was left open under ambient conditions for additional 12 h.

### 2.2. Sample Characterization

X-ray patterns were recorded on a STOE STADI MP diffractometer using MoKα radiation (λ = 0.70926 Å). The primary beam was formed using a bent Ge (111) Johann-type monochromator. The recording was carried out in a transmission mode with a thin layer of the powder sample confined between two thin films. The signal was registered using a position-sensitive DECTRIS MYTHEN detector in a 2Θ angle range of 3–50° with an acquisition time of 10 s. The phase analysis was carried out using the ICDD PDF-2 (2009) powder diffraction database. The structural data were obtained from the ICSD database.

High-resolution transmission electron microscopy (HRTEM) on a ThemisZ (Thermo Fisher Scientific, Waltham, MA, USA) electron microscope with an accelerating voltage of 200 kV and the maximum lattice resolution of 0.07 nm was used to investigate the structure of the alloys. The TEM images were recorded with a Ceta 16 (Thermo Fisher Scientific, Waltham, MA, USA) CCD matrix. Elemental mappings (EDX) were conducted using a SuperX detector (Thermo Fisher Scientific, Waltham, MA, USA). The samples for the HRTEM study were deposited on a holey carbon film mounted on a copper grid. The ultrasonic dispersal of the samples (suspensions) in ethanol was used. The morphology of particles of the powder alloys was studied using a JSM-6460 LV scanning electron microscope (Jeol, Tokyo, Japan). The specific surface area was calculated by the Brunauer–Emmett–Teller (BET, ASAP-2400 instrument, Micromeritics, Norcross, USA) method from the adsorption isotherms of argon.

The ^59^Co Internal Field NMR spectra were recorded at room temperature on a Bruker Avance 400 spectrometer outside the magnetic field of the device. The spectra were recorded using a solid-state echo pulse sequence *θ-τ-θ*, in which the pulse length *θ* was 1 μs, and the delay between pulses *τ* was 8 μs. Due to the large width of the Internal Field NMR spectra (approximately 20–50 MHz), the recording was taken point by point with a carrier frequency step of 0.5 or 1 MHz. The intensity of the spectrum at each step was determined as an integral of the Fourier transform of the absorption signal. In addition to the frequency sweep, the spectra were also recorded at various RF radiation powers to correct the intensity by the enhancement factor, which differs for various cobalt structures. The spectra were also corrected for the general frequency dependence of the intensity of the NMR spectra: I ~ ω^2^.

A T64000 Raman spectrometer (Horiba Jobin Yvon) with a micro Raman attachment was used to measure the Raman spectra. The 514.5 nm line of an Ar+ laser was used as the Raman excitation source. The spectral resolution was 1.5 cm^−1^ or better. The detector was a CCD matrix cooled with liquid nitrogen. The radiation power reaching the sample was 0.2 mW. The 520.5 cm^−1^ band of a silicon single crystal was used to calibrate the spectrometer.

## 3. Results

The structure of the mechanically alloyed powders depends greatly on the structure of the initial powders used in the synthesis, and thus, we have characterized the commercial Co and Zr powders as the first stage of the study.

### 3.1. Initial Co and Zr Metal Powders

#### 3.1.1. XRD and SEM

The diffraction pattern of the initial cobalt powder contained two sets of lines corresponding to face-centered cubic (*fcc*) (PDF Number: 00-015-806) and hexagonal close-packed (*hcp*) (PDF Number: 00-001-1278) structures (Figure 1a). The lines of the hexagonal cobalt were shifted compared to the literature data and broadened; therefore, the presence of the lines corresponding to this phase may be caused by the stacking faults in the cubic cobalt phase. The detailed analysis of the peaks gave the size of the coherently scattering regions (CSR) of approximately 21 nm (Table 2). Scanning electron micrographs demonstrated that the initial cobalt powder was represented by agglomerates of elongated rounded particles with a diameter of approximately 1 µm and a length of approximately 5 µm (Figure 2a).

According to the XRD data (Figure 1b), the initial zirconium powder contained the metallic zirconium phase (major phase, 70–80%) with a cubic structure (PDF number 00-005-0665) with a size of the coherent scattering regions of approximately 100 nm (Table 1). Additionally, minor phases were present: monoclinic zirconium oxide (PDF Number: 00-037-1484), traces of tetragonal zirconium oxide (PDF Number: 00-050-1089) and a noticeable (5–7%) amount of zirconium hydride with tetragonal structure (PDF Number: 00-034-0690). According to SEM data (Figure 2b), the zirconium powder consisted of irregular particles (ranging in size from 1 to 10 μm), close in composition to metallic zirconium, between which aggregates of very small particles with a high oxygen content were observed. Most likely, the oxide phases in the zirconium powder resulted from long-term storage in contact with atmospheric oxygen, while the hydride was formed during the synthesis of zirconium from the oxide phases by a reduction in hydrogen.

#### 3.1.2. ^59^Co Internal Field NMR

In the ^59^Co Internal Field NMR spectrum of the initial Co powder (Figure 3), one can observe three well-defined peaks at frequencies of approximately 208.5, 213.5 and 219 MHz, as well as a minor line at a frequency of approximately 216 MHz. The best-defined line located at the frequency of 213.5 MHz corresponds to multi-domain Co particles with the *fcc* crystal structure [22]. The line at a resonant frequency of approximately 216 MHz is most often attributed to the *fcc* crystal packing in small single-domain Co particles [34]. However, due to the absence of any enhancement factor distribution in this case, this line most likely corresponds to defects in cubic packing, which can formally be attributed to the *hcp* structure.

Cobalt in *hcp* packing is often observed in the spectra as a broad line stretching in the frequency range from 214 to 221 MHz [28,35]. Thus, the line at a resonant frequency of 219 MHz could be attributed to the *hcp* packing of cobalt or, equally possible, packing defects of the *fcc* crystal structure, since they are locally indistinguishable from each other.

The lines at frequencies of 208.5 MHz and 224.5 MHz are connected with the presence of iron impurities in cobalt. Jay and Wójcik [36,37] have demonstrated that the replacement of one Co atom by a Fe atom among eight nearest neighbors (NN) of the ^59^Co site in a *bcc* metal increases the local magnetic field and, consequently, the NMR resonant frequency of this site by approximately 10.7 ± 0.5 MHz. At the same time, the presence of one Fe atom among the next nearest neighbors (NNN) lowers the resonant frequency of the Co site by approximately 5 MHz. In the first approximation, for dilute samples with low Fe concentrations, both of these effects can be considered linearly additive and independent.

For the *fcc* structure of metallic cobalt, the same effect was observed in ref. [38], where, for the Co sites with three or more nearest Fe neighbors, the same linearly additive shift was observed with a step of approximately 10 MHz for each iron atom. Thus, we can assume that, in the case of the *fcc* Co structure, the presence of one Fe atom among the next nearest neighbors causes a decrease in the resonant frequency by 5 MHz similar to the *bcc* structure. Decomposition of the spectrum into Gaussian lines (the results are shown in Table 3) showed that the relative intensity of the Co [1Fe NN] line was approximately 4%, and the relative intensity of the Co [1Fe NNN] line was 3%. Such a concentration of Co [1Fe NN] sites corresponds to a relative iron content in the sample of approximately 0.35%, which may be present in the electrolytic cobalt powder PK-1u according to the manufacturer specification. It is important to note that the intensity of the Co [1Fe NNN] line should be half the intensity of the Co [1Fe NN] line, since in the *fcc* structure, the Co sites have 12 nearest neighbors and 6 next nearest neighbors, which holds true within the experimental uncertainty for the sample studied in this work.

It is also necessary to indicate the presence of a small shoulder at a frequency of approximately 210–211 MHz; however, it is rather difficult to determine the intensity of this line due to the superposition with the *fcc* Co line at 213.5 MHz. A similar line was observed in ref. [39], which was attributed to disordered cobalt structures, such as *hcp* stacking defects or grain boundaries. It does not appear possible to accurately assign this line to any particular structure. As the relative intensity of this line is less than 1%, its presence does not affect the consideration of the main lines in the spectrum.

The initial metal powders of Co and Zr were mixed at an atomic ratio of 53:47 and subjected to MA. According to the XRD measurements, the MA product contained metallic cobalt of two structural types, similar to cobalt in the original powder (Figure 1a and Figure 4a). Judging by the ratio of peaks 111 for Co(*fcc*) and 100 and 101 for “Co(*hcp*)”, the proportion of “hexagonal” (or defective cubic) cobalt has decreased after MA.

### 3.2. Products of Mechanical Alloying

#### 3.2.1. XRD and SEM

Zr oxides and metallic Co were the major phases detected by XRD in the samples after MA. Peaks of *fcc* and *hcp* Co were detected (the same peaks were detected in the pattern of the initial Co powder), which indicates the presence of stacking faults in the particles of metallic Co. Zirconium oxides are present in two modifications, monoclinic and tetragonal. All identified phases were characterized by extremely small coherent scattering region sizes (3–9 nm) as compared with the CSR sizes of the phases of the initial powders (Table 2). Trace amounts of Co_3_O_4_ spinel (PDF Number: 01-078-5622) were also detected.

In addition to lines related to the zirconium oxide phases, there were other peaks that could be attributed either to ZrH_2_ phase, as in the initial Zr powder, or to the defect Zr phases. However, the lattice parameters of the ZrH_x_ phase in the Co-Zr sample [a = 4.72 (1), c = 4.95 (1)] were found to be quite different from those of the initial ZrH_x_ [a = 4.66(1), c = 4.79 (1)] (Table 2). Thus, we have denoted this phase as Zr *. The ratio between the intensities of the peaks corresponding to Zr oxides and metallic Zr indicated severe oxidation of zirconium during the MA process (Figure 1b and Figure 4a).

All identified phases were characterized by extremely small coherent scattering region sizes within 3–9 nm compared to the CSR sizes in the initial powders (Table 2). No Co-Zr alloys were found. Probably, this was the result of poor contact surface between metal particles of Co and Zr [40].

According to the SEM analysis (Figure 5), the products of MA contained aggregated particles varying from several tens of microns to less than one micron.

#### 3.2.2. HRTEM and EDX Mapping

Figure 6 demonstrates the typical results of EDX elemental mapping obtained for the mechanically alloyed Co-Zr sample. The alloy consisted of agglomerates with block structure. The primary crystallites of this structure ranged from 5 to 200 nm. According to the EDX measurements, oxygen was predominantly located in the surface layers of the particles, forming an oxide shell containing both cobalt and zirconium oxides. The zirconium oxide was represented by 20–100 nm crystallites of a cubic phase with tetragonal distortions (Figure 7). Some monoclinic ZrO_2_ crystallites were also present in the samples. The cobalt oxide phase was represented by much smaller crystallites of Co_3_O_4_ spinel (5–20 nm). Metallic particle cores were clearly visible in the HRTEM images of the MA products with the Co metal present in the form of crystallites more than 100 nm in size (Figure 7a). The Zr metal was represented by areas with large microstrains that manifested themselves as the Moiré patterns in the TEM image (Figure 7b). The presence of these microstrains may also be connected with the presence of ill-defined reflection peaks in the XRD patterns that were attributed to the highly defective metallic Zr * phase.

The inhomogeneity of the MA Co-Zr product was also confirmed by the Raman spectroscopy data (Figure 8). The diameter of the laser beam used in the setup was approximately 2 µm, which allowed observing different parts of the sample. Different points in the sample demonstrated the prevalence of either zirconium oxide (143, 245, 301, 435 and 625 cm^−1^ bands corresponding to vibrational modes of cubic ZrO_2_ [41]) or cobalt oxide with spinel structure and vibrational modes of 190, 470, 511, 606, and 674 cm^−1^. All these bands correspond to F_2g_, E_g_, F_2g_, F_2g_, A_1g_ Raman bands of Co_3_O_4_ [42].

#### 3.2.3. ^59^Co Internal Field NMR

The ^59^Co Internal Field NMR spectra of the mechanically alloyed products are shown in Figure 9. In contrast to the spectrum of the original metallic Co, only two strongly broadened lines could be distinguished in these spectra, related to large multi-domain particles of metallic cobalt with *fcc* (approximate position 213 MHz) and *hcp* (approximate position 218–220 MHz) structures. Similarly to the case of the initial metal, the *fcc* line of Co can unambiguously be assigned, while the broad shoulder, formally related to *hcp* packing, can originate from both the *hcp* Co phase and defects in the *fcc* phase.

In addition to the lines of metallic cobalt, each spectrum demonstrated a shoulder extended to lower frequencies (observed at resonant frequencies of 190–210 MHz). As mentioned above, according to the literature data, the presence of one Zr atom in the local environment of the Co atom leads to a decrease in the resonant frequency by approximately 30 MHz [31,32]. Thus, a line from the Co [1Zr NN] should be observed at a frequency of approximately 183–186 MHz, but the intensity in this part of the spectra was zero. In ref. [43], the appearance of a shoulder extended to low frequencies was associated with an increase in the number of grain boundaries, which occurred due to particle fracturing during mechanical activation of pure metallic cobalt. In the same work, a similar change in the ratio of the *fcc* and *hcp* lines, associated with an increase in the defect concentration in the particles, was observed. Thus, according to the ^59^Co NMR spectroscopy data, ball-milling treatment led to a significant increase in the defect concentration in the metallic cobalt particles and the formation of additional grain boundaries characterized by a weaker magnetic order.

The apparent absence of the Co sites with neighboring Fe atoms may be caused by general amorphization of the particles and partial oxidation by water, which was shown to occur by XRD and SEM. Additionally, ^59^Co NMR data did not reflect the formation of magnetic alloys or solid solutions of Co and Zr; however, it should be remembered that the Internal Field NMR allows investigation of exclusively ferromagnetic phases, while mechanical alloying of Co and Zr powders can produce zirconium-rich paramagnetic phases.

The magnetic structure of the studied cobalt particles has a significant effect on the ^59^Co Internal Field NMR spectra. Thus, in sufficiently large particles divided into magnetic domains (multi-domain particles), the latter are separated by domain walls, the movement of which, under the action of external radio frequency radiation, leads to a significant signal amplification compared to the domains themselves. If the particle size is less than favorable for division into domains (approximately 70 nm [44]), the particle is single-domain, i.e., the NMR signal enhancement from such a particle is by 1–2 orders of magnitude weaker compared to domain walls. Thus, the ^59^Co Internal Field NMR spectra are recorded with a sweep in the power of the incident RF radiation to account for the difference in the enhancement factors between domains and domain walls. Since no significant differences in the signal enhancement were observed either in the initial Co powder or in the Co-Zr products after MA over the entire frequency range, we could conclude that there were no single-domain particles in these samples.

### 3.3. Hydrogen Treated Sample

#### 3.3.1. XRD and SEM

According to the X-ray diffraction patterns (Figure 4b) the samples did not undergo any noticeable changes after treatment with high-pressure hydrogen. Zirconium oxides and Zr * remained the dominating phases in all samples, while cobalt was present in the oxidized form and metallic form as *fcc* phase of defect structure. A significant increase in the amount of small particles after the hydrogen treatment observed by SEM demonstrated the effect of hydrogen pulverization on the fineness of the obtained powders (Figure 10). Nevertheless, the sizes of the coherent scattering regions remained in the range of 4–9 nm.

#### 3.3.2. ^59^Co Internal Field NMR

The ^59^Co IF NMR spectrum of the hydrogen treated sample is shown in Figure 9 (blue). This spectrum was characterized by a decrease in the relative intensity of the line at 213.5 MHz corresponding to the multi-domain *fcc* structure and the appearance of a line with a resonant frequency of approximately 216.5 MHz, which is characteristic for single-domain particles of metallic cobalt with *fcc* crystal structure. The presence of small single-domain nanoparticles in the sample is evident from the distribution of the optimal magnetic field strength, which is inversely proportional to the enhancement factor (Appendix A). There, the curve of the optimal magnetic field is divided into parts corresponding to multidomain particles (red, low optimal field → high enhancement factor) and single-domain particles (light green, high optimal field → low enhancement factor). 

At the same time, there was a significant increase in the intensity of Co [1Fe NN] and Co [1Fe NNN] sites (Table 2), which were practically absent in the MA product. The reason for this may be the reduction of iron (and, possibly, adjacent cobalt) during treatment with hydrogen under high pressure. At the same time, the total content of iron in the products of MA and after treatment with hydrogen remained small, since the data given in Table 3 relate only to metallic cobalt in the samples without considering zirconium.

## 4. Discussion

### 4.1. “Fukushima” Effect

The initial powders of metallic Co and Zr were represented by large micron-sized agglomerated particles consisting mainly of the corresponding metal phases with some evidence of impurities and oxidation. As evidenced by the ^59^Co Internal Field NMR technique, the electrolytic cobalt contained a small amount (<0.5%) of iron as an evenly distributed solution, while the main impurities in the Zr powder were Zr hydride and oxide, most probably formed during the synthesis and storage of the powder, respectively. Zirconium is a highly reactive metal that cannot be stored in open air. In our case, the Zr powder was stored in water. This method of storage (the residual moisture in the Zr powder) had a significant impact on the synthesis outcome.

The MA processing resulted in partial amorphization of cobalt particles and introduced a large amount of stacking faults, as evidenced both by XRD and 59Co IF NMR. The sizes of CSR of different phases decreased by an order of magnitude. The IF NMR spectra demonstrated the presence of stacking faults in the metallic Co phase. However, the most significant changes were observed for the zirconium powder, which experienced severe oxidation. Additionally, a part of Zr was present in the form of very defective Zr. The oxidation of metallic Zr was due to the following reaction:$$\mathrm{Zr}+2H_{2}O\;\to \;{\mathrm{ZrO}}_{2}+2H_{2}+6.5\;\mathrm{kJ}/g$$

Oxidation of zirconium with water vapors is the so-called parazirconium reaction, which caused the hydrogen explosions during the Fukushima nuclear power plant disaster in 2011. This reaction takes place only at elevated temperatures (higher than 900 °C). The average temperature of balls in planetary mills of AGO-2 type is less than 600 °C [45,46]. However, local temperature in the points of ball collision may be much higher (“Magma-Plasma Model” [47]). At the same time, no alloying between Co and Zr was detected under experimental conditions selected in the present work.

### 4.2. Formation of Single-Domain Cobalt Particles during Hydrogenation

The high-pressure treatment with hydrogen at room temperature was applied for the purpose of increasing the fineness of the powders obtained by MA. For evaluating the effect of this treatment, the ^59^Co IF NMR has proven to be the most efficient technique since it was able to detect the presence of single-domain metallic Co particles. The value of the particle size, at which the existence of only one magnetic domain becomes energetically favorable, is usually given as ~70 nm. According to the NMR data, the hydrogen-treated samples contain a large number of particles with a diameter smaller than this boundary value. Most often, such a resonance line of single-domain particles with fcc packing was observed for deposited samples with a relatively low content of cobalt. For the first time, this resonance was observed for particles 10–15 nm in size deposited on γ-Al_2_O_3_ [34]. This line was also observed both for particles with a size of 35–50 nm deposited on SiC [35] and for very small particles (with a size of approximately 4 nm) placed in carbon nanotubes [28]. Thus, the sizes of the particles giving a signal at this frequency lie in the range of 5–50 nm, proving that low-temperature high-pressure hydrogen treatment indeed increases the fineness of particles.

According to ^59^Co Internal Field NMR, high-pressure hydrogenation at room temperature leads to the formation of “single-domain” particles of metallic cobalt, which have sizes less than 70 nm. As shown by the XRD measurements, the average CSR size did not change significantly after the hydrogen treatment (4–6 nm both after mechanical alloying and hydrogen treatment). This seeming discrepancy has the following explanation. According to both techniques, the MA process introduces a large amount of stacking defects into the structure of the Co particles, which is evidenced by an increase in the “*hcp*” signal in the IF NMR spectra. Nevertheless, the connectivity between the crystallites in the mechanically alloyed products remains such that only multi-domain particles (larger than 70 nm) were observed in the IF NMR spectra. The subsequent treatment with hydrogen results in pulverization of the agglomerated particles owing to the presence of weak points, possibly introduced during MA by mechanical collisions or the chemical reaction of Zr with water vapor. Thus, the physical connectivity between the Co crystallites breaks, which results in the loss of magnetic exchange connectivity and appearance of the single-domain particle signals in the IF NMR spectra. A possible explanation for the appearance of these particles is that the initial metallic Co is represented by crystallite agglomerates. MA followed by high-pressure hydrogen treatment leads to weakening of the bonds between these crystallites, breaking them down into smaller particles (Figure 11). At the same time, the average CSR size does not change, which is evidenced by XRD. Additionally, this scheme allows the explanation of a significant decrease in the relative content of the *hcp* Co structure, from 67% to 33%, (Table 2) by a decrease in the concentration of volumetric extended defects. An increase in the average dispersion after hydrogenation of Co-Zr composite was also evidenced by an increase in its specific surface area from 4.5 m^2^/g after MA to 9.5 m^2^/g after hydrogenation.

### 4.3. Metal-Oxide Nanocomposite as a Precursor for the Fischer–Tropsch Catalyst

This work has shown that no alloying occurred in the Co-Zr materials during MA or high-pressure hydrogen treatment. Nevertheless, this does not mean that the obtained metal oxide composite cannot demonstrate activity in the heterogeneous catalytic processes. The Co-Zr catalysts obtained by co-precipitation [48] or impregnation [49] techniques and containing Zr in the oxidized form were employed in Fischer–Tropsch synthesis. In those works, *hcp* Co was demonstrated to be the most active metallic Co packing. In our work, we formed nanoparticles of Co supported on the surface of zirconium oxide particles, which undergo partial oxidation under ambient atmosphere. Thus, we can suppose that this metal oxide composite may demonstrate high activity in Fischer–Tropsch synthesis after reductive pre-treatment, which will be the subject of further study. The mechanical alloying leads to an increase in hcp Co sites, while hydrogen pulverization leads to an increase in dispersion of cobalt particles. Both these factors can significantly improve activity of the treated powders in FTS.

## 5. Conclusions

For the first time, mechanical alloying followed by high-pressure hydrogen treatment was proposed as a route for the synthesis of Co-Zr materials with small particle sizes. These alloys can be used as heterogeneous catalysts or gas absorption materials. Co-Zr powder mixtures with starting Co:Zr ratios of ~1:1 were mechanically alloyed and subsequently treated with hydrogen at room temperature and a pressure of 5.0 MPa. The results of X-ray diffraction and ^59^Co Internal Field NMR (^59^Co IF NMR) have demonstrated that the mechanical alloying stage introduces a large amount of defects and stacking faults into the structure of metallic Co particles, leading to a drastic decrease in the coherent scattering region sizes. An increase in the concentration of hexagonal Co was observed. At the same time, the defect metallic Zr was formed. The particles were partially oxidized forming ZrO_2_ due to the “Fukushima” effect, caused by the presence of residual water, in which the Zr powder was stored. No alloying between Co and Zr was evidenced by any of the employed techniques.

According to the ^59^Co IF NMR measurements, high-pressure hydrogen treatment led to the appearance of small single-domain metallic Co nanoparticles smaller than 70 nm in size. At the same time, the XRD measurement did not reveal any change in the CSR size of the samples. Thus, the high-pressure hydrogen treatment led to pulverization of Co particles without introducing any changes to their crystalline structure, which makes this material interesting for application in catalysis. The work concerning the application of Co-Zr nanocomposites as active components in Fischer–Tropsch synthesis is currently in preparation.

## Figures and Tables

**Figure 1 materials-16-01074-f001:**
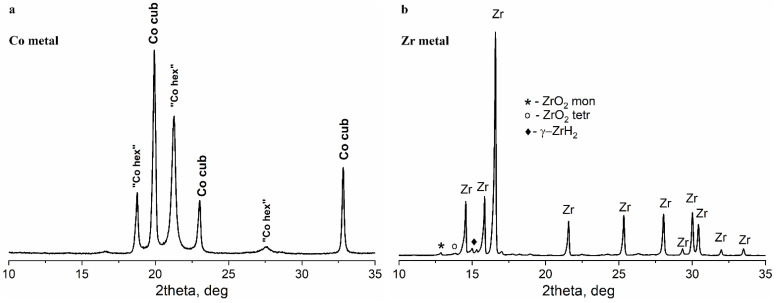
(**a**) X-ray diffraction pattern of the initial metallic cobalt powder; (**b**) X-ray diffraction pattern of the initial metal zirconium powder. The diffraction lines corresponding to minor phases in the sample are marked with symbols.

**Figure 2 materials-16-01074-f002:**
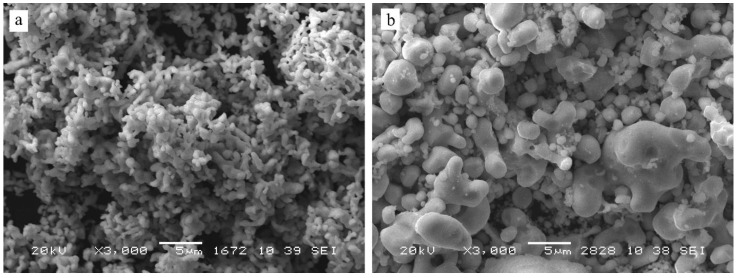
Scanning electron micrographs of the (**a**) initial Co and (**b**) initial Zr powders.

**Figure 3 materials-16-01074-f003:**
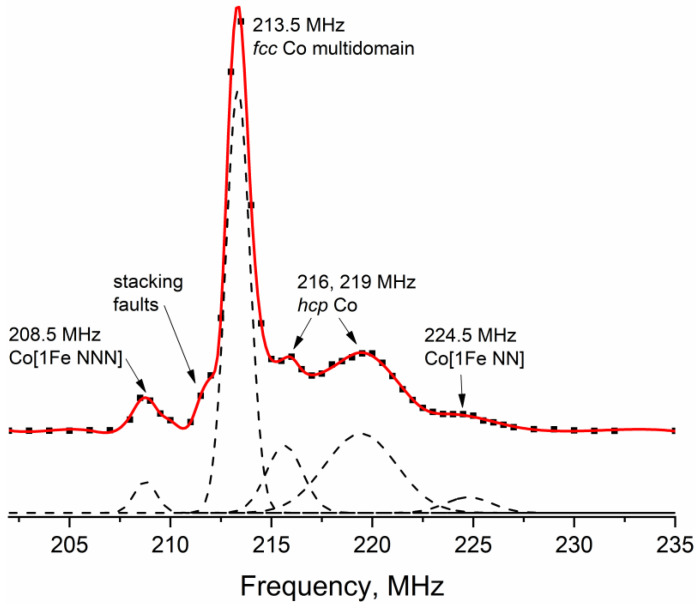
^59^Co Internal Field NMR spectrum of the initial Co powder. Experimental points are shown with black squares; interpolation curve is shown in red. Dashed line shows the theoretical Gaussian lines used in the spectral decomposition.

**Figure 4 materials-16-01074-f004:**
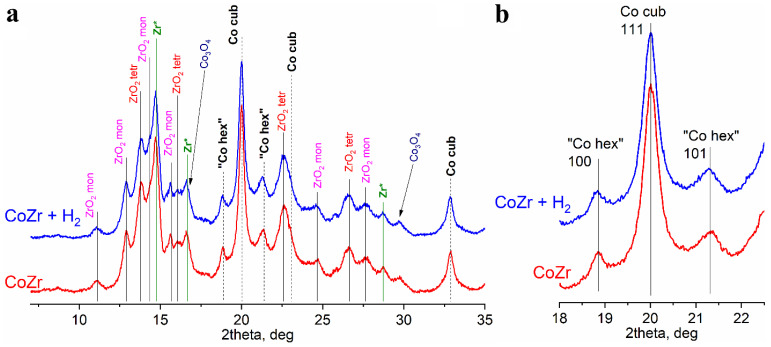
(**a**) XRD patterns of the mechanically alloyed (red) and subsequently hydrogen treated (blue) mixtures of Co and Zr powders, shifted along the *y*-axis; (**b**) fragments of the XRD diffraction patterns showing the main peaks related to Co with *fcc* structure (Co cub) and Co with *hcp* packing (“Co hex”).

**Figure 5 materials-16-01074-f005:**
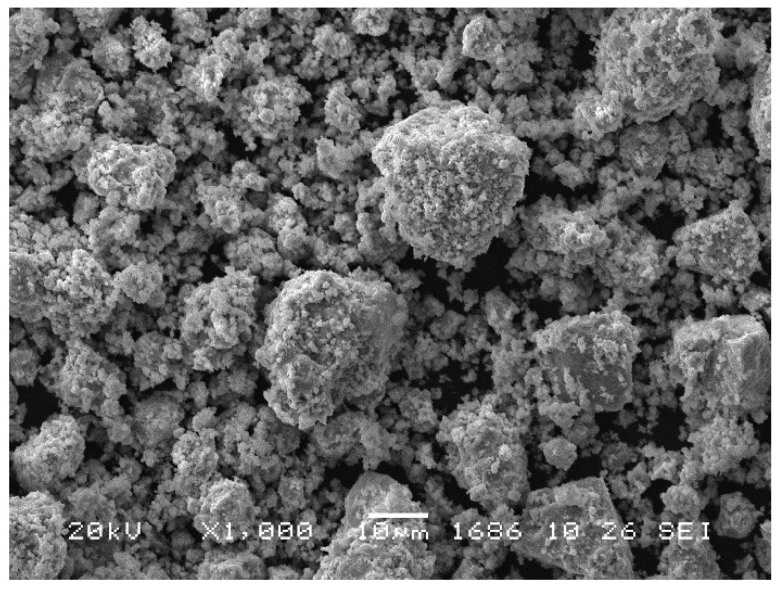
Scanning electron micrographs of the mechanically alloyed Co-Zr sample.

**Figure 6 materials-16-01074-f006:**
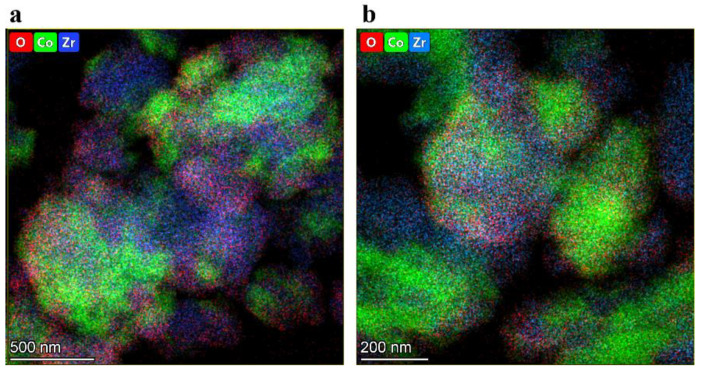
Typical EDX mapping of the Co-Zr product: (**a**) after mechanical alloying; (**b**) after mechanical alloying and hydrogen treatment.

**Figure 7 materials-16-01074-f007:**
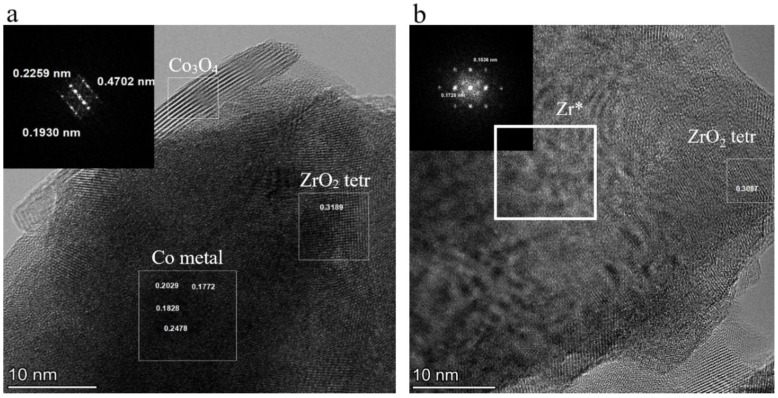
HRTEM images of: (**a**) typical Co-Zr mechanical alloying product particle with a metallic Co core. Top left insert-FFT image from Co_3_O_4_ area; (**b**) typical Co-Zr mechanical alloying product particle with a metallic Zr core. Top left insert-FFT image from Zr * area with microstrains.

**Figure 8 materials-16-01074-f008:**
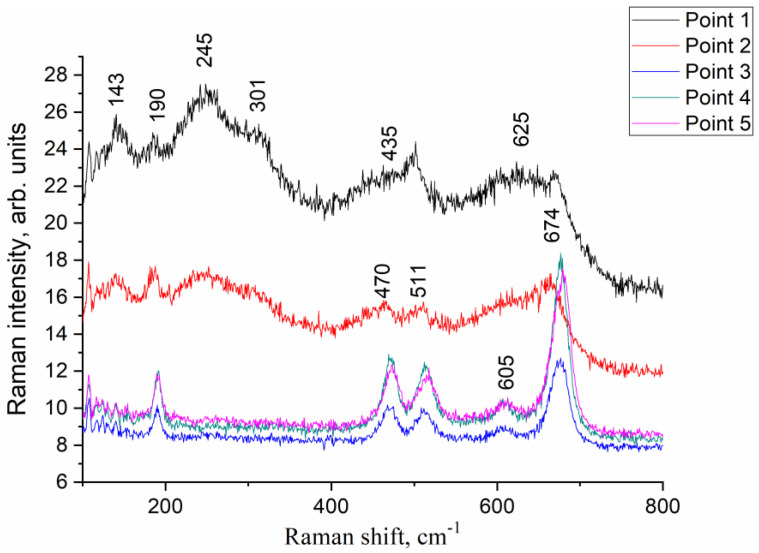
Raman spectra of Co-Zr mechanical alloying product from different points in the sample: for points 3–5, only bands of Co_3_O_4_ spinel are present; for points 1–2, additional bands of ZrO_2_ are observed.

**Figure 9 materials-16-01074-f009:**
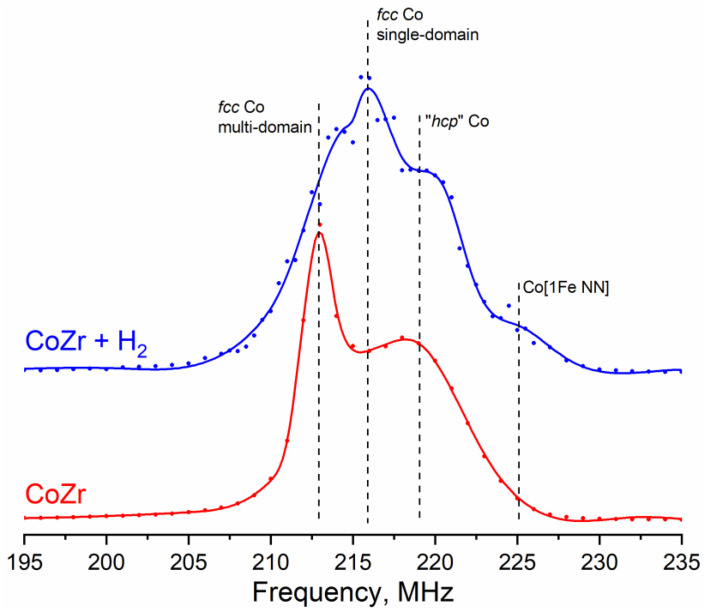
^59^Co Internal Field NMR spectra of the Co-Zr mechanical alloying product (red) and the product of hydrogen treatment (blue). Experimental points are shown with symbols, solid lines show the interpolation curves. Approximate positions of signals corresponding to different cobalt structures are marked with dashed lines.

**Figure 10 materials-16-01074-f010:**
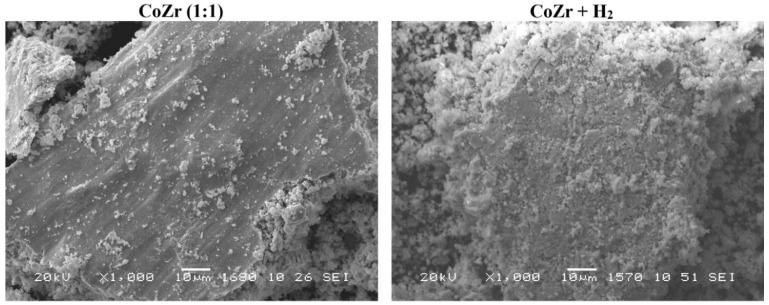
SEM images of the Co-Zr mechanical alloying product particles in contact with the ball mill shell before (**left**) and after (**right**) high-pressure hydrogen treatment.

**Figure 11 materials-16-01074-f011:**
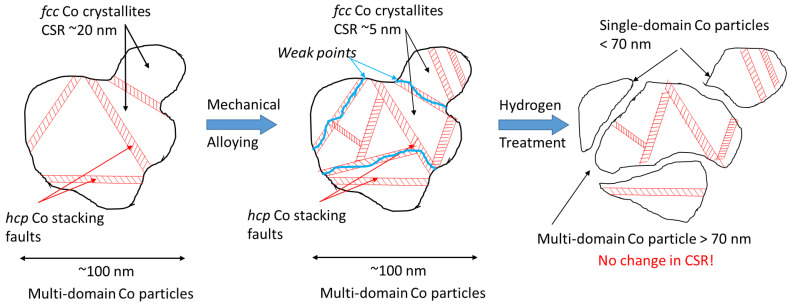
Possible scheme describing the effect of mechanical alloying and hydrogen treatment on the metallic Co particles. Mechanical alloying introduces high amounts of stacking faults and weak points into the initially well-ordered particles, decreasing the average size of the coherent scattering regions (CSR). The subsequent hydrogen treatment breaks the particles at weak points, forming magnetically single-domain particles without changing the CSR.

**Table 1 materials-16-01074-t001:** Structural parameters of Co-Zr powders after mechanical alloying.

Co:Zr ratio, at.%	Mechanical Alloying Conditions	Milling Time, h	Heat Treatment	Phases	Ref.
34:66	High energy planetary mill NARYA 250 Amin Co. Ar 3 atm, ball:powder ratio 15:1. 400 rpm.	2 24 42	No No No	Co, Zr Amorphous phases, Zr_3_Co Amorphous phases	[13]
82:18	High energy planetary mill, ball:powder ratio 30:1, toluene medium, 6 h at 200 rpm.	8	Preliminary melting and annealing at 1000° then crushed	Main phase Zr_2_Co_11,_ minor phase Zr_6_Co_23_	[15]
32:68	Planetary ball mill (Retsch PM 400), Ar, ball:powder ratio 15:1. 300 rpm	2 8 16	No No No	Zr, Co, ZrO_2_ Zr, Co, ZrO_2_, Zr_3_Co Amorphous phases	[14]
33:67 (intermetallic) 50:50 (intermetallic)	High energy planetary mill, Ar, ball:powder ratio 10:1. 300 m s^−2^ 600 m s^−2^ 600 m s^−2^ 900 m s^−2^	Unknown	No No No No	Amorphous phases ZrCo, Zr_3_Co Amorphous phases Zr_2_Co, Zr_3_Co	[16]

**Table 2 materials-16-01074-t002:** Structural parameters and phase compositions of the initial Co and Zr powders.

Sample	Parameter	Phase				
		Co (*fcc*, *hcp*)	ZrO_2_ Monoclinic	ZrO_2_ Tetrahedral	ZrH_2_	Zr
Co	CSR (nm)	21(1)				
	wt%	100				
Zr	CSR (nm)		≈30	-	≈30	≈100
	wt%		11	5	7	77
	Lattice parameter (Å)		a = 5.23(1) b = 5.10(1) c = 5.34(1) β = 99.3(1)	-	a = 4.661) c = 4.79(1)	a = 3.231(1) c = 5.145(1)

**Table 3 materials-16-01074-t003:** Relative contributions of different structures of metallic Co according to decomposition of the experimental ^59^Co IF NMR spectra.

Sample	Co Structures Relative Contribution, %				
	Co [1Fe NNN] 208.5 MHz	*fcc* Co m. d. 213.5 MHz	*fcc* Co s. d. 216.5 MHz	*hcp* Co 214–221 MHz	Co [1Fe NN] 224.5 MHz
Co	3	52		41	4
CoZr		33		67	
CoZr + H_2_	2	26	32	36	4

## Data Availability

The data presented in this study are available on request.

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
