# Peer review of "Formation of Metal-Oxide Nanocomposites with Highly Dispersed Co Particles from a Co-Zr Powder Blend by Mechanical Alloying and Hydrogen Treatment"

_materials, 2023, doi:10.3390/ma16031074_

Round 1
Reviewer 1 Report
In this study, the authors investigated the effect of mechanical alloying and subsequent low- temperature high-pressure hydrogen pulverization on the structural and spectral properties of mixtures of cobalt and zirconium powders. The results are technologically sound and can provide some interesting insights into the mechanical treatment of powdered metal blends for using in various fields. Before some issues are still needed to be clarified before the manuscript can be accepted:
1. The authors claimed that hydrogen pulverization induced particles with size of ~70 nm. However, it seems that only Internal Field NMR results support such claim. Can the authors provide more direct evidence, such as SEM and TEM images to show the real particle size?
2. Can the treatment employed by the authors in the present study enhance available metallic cobalt sites in the catalyst for Fischer-Tropsch synthesis? I hope the authors can provide more discussions.
Author Response
- Can the authors provide more direct evidence, such as SEM and TEM images to show the real particle size?
Significant increase of the amount of small particles after hydrogen treatment was observed in the SEM image (Fig 10).
The sentence describing Fig. 10 (lines 361 - 363) was revised from:
Additional macropores were detected at the sites of contact between the particles and the ball mill shell that may indirectly demonstrate the effect of hydrogen pulverization on the fineness of the obtained powders (Fig. 10).
To (lines 363 - 366):
A significant increase of the amount of small particles after the hydrogen treatment observed by SEM demonstrated the effect of hydrogen pulverization on the fineness of the obtained powders (Fig. 10).
This conclusion was also supported by an increase of specific surface area after hydrogenation.
- Can the treatment employed by the authors in the present study enhance available metallic cobalt sites in the catalyst for Fischer-Tropsch synthesis?
A sentence was added in the Discussion (lines 472-474):
The mechanical alloying leads to an increase in hcp Co sites, while hydrogen pulverization leads to an increase in dispersion of cobalt particles. Both these factors can significantly improve activity of the treated powders in FTS.
Thank You for gentle comments!
Reviewer 2 Report
Ilya Yakovlev et al mentioned about the formation of metal-oxide nanocomposites from a powdered Co-Zr blend using mechanical alloying and hydrogen treatment. Overall, based the data provided, it can be considered for publication in this journal. However, there are some issues that have to be fixed before publication;
The title should be revised to somewhat catchy type.
The abstract could be more specific to increase the interest of readers.
Please revised introduction with proper consequences with some new references by exploring the literature. Also mention some composite which were synthesized by doping chemically (Journal of alloys and compounds 578, 431-438, Applied Catalysis A: General 505, 507-514).
Should mention , why author choose this material.
Please specify the novelty of the study.
The figure should be revised to high quality as some are in blur format.
Some errors regarding the sub/super script, spacing and typo need to consider throughout the manuscript.
Make sure that the format of references are uniform.
In the conclusion author mention that this material will be suitable to application however did not mention application.
Author Response
The title should be revised to somewhat catchy type.
We agree. The title was changed to:
Formation of metal-oxide nanocomposites with highly dispersed Co particles from a powdered Co-Zr blend after mechanical alloying and hydrogen treatment
The abstract could be more specific to increase the interest of readers.
A sentence (line 23) was added to the abstract:
Moreover, an increase of hcp Co sites was observed.
Sentence in lines 23-24 was changed from:
Such pulverization of Co made this nanocomposite suitable for the synthesis of promising Fischer-Tropsch catalysts.
To:
Such pulverization of Co and increase of hcp Co sites made this nanocomposite suitable for the synthesis of promising Fischer-Tropsch catalysts.
Please revised introduction with proper consequences with some new references by exploring the literature. Also mention some composite which were synthesized by doping chemically (Journal of alloys and compounds 578, 431-438, Applied Catalysis A: General 505, 507-514).
Unfortunately, both references suggested by the Reviewer fall out of the rather narrow scope of our study (mechanically alloyed Co-Zr blends) in both the sample composition and the preparation technique. Thus, we do not believe that it is possible to include them into our discussion.
Should mention , why author choose this material.
As we have stated in the Introduction section of the article, zirconium-containing alloys are promising compounds for heterogeneous catalytic processes (among which Fischer-Tropsch synthesis is of particular interest) according to vast literature data. Additionally, alloys can provide higher activity per unit volume of the reactor due to higher density compared to conventional oxide catalysts.
Please specify the novelty of the study.
To our best knowledge, this is the first work concerning the hydrogen treatment of the mechanically alloyed Co-Zr blends.
We modified Abstract: In this work, for the first time metal-oxide nanocomposite powder was produced by mechanical alloying (MA) in a high-energy planetary ball mill from commercial powders of Zr and Co in the atomic ratio Co:Zr = 53:47 in inert atmosphere followed by high-pressure hydrogenation at room temperature.
Also we modified Conclusions: For the first time consequent mechanical alloying and high-pressure hydrogen treatment has been proposed as a route for the synthesis of Co-Zr blends with increased particle fineness that can be later used as heterogeneous catalysts or gas absorption materials. Co-Zr powder mixtures with starting Co:Zr ratios (~ 1:1) have been mechanically alloyed and subsequently treated with hydrogen at room temperature and 5.0 MPa pressure.
The figure should be revised to high quality as some are in blur format.
All of the figures were updated to at least 600 dpi resolution.
Some errors regarding the sub/super script, spacing and typo need to consider throughout the manuscript.
Formatting errors were fixed throughout the manuscript.
Make sure that the format of references are uniform.
The references were formatted according to the MDPI standards.
In the conclusion author mention that this material will be suitable to application however did not mention application.
We agree. We have added the next sentence at the end of the Conclusions section: The work concerning the application of Co-Zr nanocomposites as active components in Fischer-Tropsch synthesis is currently in preparation.
Thank You very much for useful comments.

Reviewer 3 Report
The manuscript entitled “Formation of metal-oxide nanocomposites from a powdered Co-Zr blend after mechanical alloying and hydrogen treatment”. Some issues to be addressed will improve the quality of the manuscript. Therefore, I recommend this work could be published after the major revision
1. Should the author write down the novelty of this article in the abstract?
2. The English composition requires many improvements. The authors should proofread the manuscript carefully to minimize grammatical errors.
3. All the references mentioned in the paper should be cited in the text or vice-versa.
4. The Co-Zr blend after mechanical alloying has been widely studied, and many studies have been performed. The author, please add a comparative table for the reader's clear understanding.
5. To improve the quality of the manuscript, I suggest authors read the recommended manuscript carefully and cite it in its proper place.
Journal of King Saud University – Science Volume 32, Issue 4, June 2020, Pages 2397-2405; The Physics of Metals and Metallography volume 107, pages478–483 (2009)

Author Response
- Should the author write down the novelty of this article in the abstract?
We agree. We revised the two sentences in the Abstract:
In this work, for the first time metal-oxide nanocomposite powder was produced by mechanical alloying (MA) in a high-energy planetary ball mill from commercial powders of Zr and Co in the atomic ratio Co:Zr = 53:47 in inert atmosphere followed by high-pressure hydrogenation at room temperature.
59Cо IF NMR spectra revealed the formation of magnetically single-domain cobalt particles after hydrogenation while the crystallite sizes remained unchanged which was not observed earlier
- The English composition requires many improvements. The authors should proofread the manuscript carefully to minimize grammatical errors.
We agree. The English in the manuscript was revised.
- All the references mentioned in the paper should be cited in the text or vice-versa.
All of the references listed in the References section are now cited in the main text of the manuscript and vice versa.
- The Co-Zr blend after mechanical alloying has been widely studied, and many studies have been performed. The author, please add a comparative table for the reader's clear understanding.
We agree. We have added such table (Table 1) into Introduction.
- To improve the quality of the manuscript, I suggest authors read the recommended manuscript carefully and cite it in its proper place.
Unfortunately, we believe that the first reference suggested by the Reviewer falls too far out of scope of our work in both the system composition and the sample preparation. Thus, we do not see a way of including it in the discussion.
The second reference suggested by the Reviewer was added in the Results section (lines 288-290)
We have added in the Results the sentences No CoZr alloys were found. Probably, this is the result of poor contact surface between metal particles of Co and Zr [40].

Reviewer 4 Report
In this work performed by Yakovlev and coworkers, Zr-Co composite was prepared by mechanical alloying followed by hydrogenation. The prepared materials were well-characterized by XRD, microscopy, internal field NMR, and Raman, and the conclusions were supported by the data. Furthermore, the authors propose a potential application side of the composite, which could be an interesting approach for expanding the formulation of FTS catalysts. This manuscript can be published after addressing the following comments.
There seem to be some misunderstanding between the term ‘absorption’ and ‘adsorption’, which are two different concepts, and the former has appeared multiple times in the manuscript. Please double-check all the places in the text and make sure the correct term is used.
As mentioned in the abstract and the introduction, one of the motivations of high-pressure hydrogenation is to increase the particle fineness, in addition to the morphological comparison as shown in Figure 10, authors need to provide the surface areas data of the sample before and after hydrogen treatment to demonstrate how much the surface area has increased after hydrogenation.
Author Response
There seem to be some misunderstanding between the term ‘absorption’ and ‘adsorption’, which are two different concepts, and the former has appeared multiple times in the manuscript. Please double-check all the places in the text and make sure the correct term is used.
We agree. We improved sentence in the Introduction (lines 32-33): Cobalt-zirconium alloys are widely used as hydrogen storage materials, active gas adsorbents under high vacuum conditions [1–9], and magnetic materials
As mentioned in the abstract and the introduction, one of the motivations of high-pressure hydrogenation is to increase the particle fineness, in addition to the morphological comparison as shown in Figure 10, authors need to provide the surface areas data of the sample before and after hydrogen treatment to demonstrate how much the surface area has increased after hydrogenation.
We agree. This data was included into Discussion and Experimental sections. According to the BET adsorption measurements, the specific surface area of the hydrogen-treated sample increased twofold compared to the mechanically-alloyed one.
